# Integrating the Expression and Discrimination via Bilateral Compensation for Molecular Property Prediction

## Abstract

Predicting molecular properties plays an important role in both scientific research and industrial applications. Given that different molecular properties are influenced by specific atoms or functional groups, it is essential to incorporate both types of information. Previous approaches either leverage subgraph information in self-supervised learning to pre-train atom-based architectures or develop subgraph-based architectures tailored to specific downstream tasks. However, these methods often lack a thorough analysis or theoretical support concerning the expressive capabilities of these two types of representations. Moreover, they typically rely on fixed coupling representations, which cannot adaptively prioritize more discriminative information for various downstream tasks. In this paper, we introduce a Route-guided Bilateral Compensation (RBC) architecture that explicitly extracts atom-wise and subgraph-wise information through two decoupled branches and integrates them via a route module. Theoretically, we demonstrate that our decomposition-polymerization subgraph-wise branch exhibits greater expressive power than the atom-wise branch, and that the integration process reduces the generalization error bound. Furthermore, we propose a coordinated self-supervised learning strategy that incorporates node-level masked graph reconstruction tasks for atomic and lexicalized subgraph tokens, alongside a graph-level contrastive learning task. For different downstream tasks, the route module facilitates dynamic integration, enhancing the discriminative power of the final representation. External experiments verify the effectiveness of our method.

## 1 Introduction

Molecular properties prediction plays a fundamental role in many tasks like drug and material discovery (Feinberg et al., 2018). Previous methods typically model molecules as graphs, where atoms and chemical bonds are modeled as nodes and edges, respectively. Graph Neural Networks (GNNs) (Hamilton et al., 2017) have been widely applied to predict specific properties associated with atoms, such as solubility and reactivity (Zhang et al., 2021; Hu et al., 2019; Yang et al., 2022). However, not all molecular properties are determined by individual atoms, and some chemical properties are closely related to functional groups (subgraphs) (Zhong et al., 2024; Li et al., 2024; Kong et al., 2022), such as efficacy, and metabolic properties. Therefore, how to fuse atom-wise information and subgraph-wise information is vital for molecular property prediction.

Existing methods to employ both kinds of information for making final predictions can be broadly grouped into two categories. One category involves utilizing subgraph-wise knowledge to pre-train atom-based architectures in the self-supervised learning stage. For instance, MGSSL (Zhang et al., 2021) integrates atom-wise self-supervised tasks with subgraph-wise tasks, such as masking and predicting subgraph tokens, achieved by transforming molecular graphs into tree structures. The other category focuses on constructing joint architectures of atoms and subgraphs for downstream tasks. For instance, in (Yang et al., 2022), the output representation is amalgamated using both atom-wise and subgraph-wise information through an attention mechanism, which is designed for downstream tasks specifically.

Achieving great success, the above two categories of methods lack theoretical analysis on the expression capacities (Wollschläger et al., 2024) of atom-wise and subgraph-wise representations, as well as the benefit of integrating them. Besides, these methods generally obtain coupling integrated representations, i.e., the mechanism by which the two types of information are integrated is fixed and the two types of information are coupled together within the final representations. However, the discriminative information for different downstream tasks is different. Transferring such statically fused representations to different tasks in a unified manner will affect their discriminative ability.

In this paper, we propose a novel Route-guided Bilateral Compensation (RBC) architecture, which consists of an atom-wise branch, and a subgraph-wise branch to explicitly decouple the two types of information. For the subgraph-wise branch, we propose a decomposition-polymerization network that interacts between atoms and subgraphs in decomposition layers and interacts among subgraph nodes in polymerization layers. A Route-guided Module is introduced to enable the dynamic selection of atom-wise or subgraph-wise information, catering to the specific requirements of molecular properties. In theory, we verify that subgraph-wise information can boost the expression of graphs, which complements the shortcomings of atom-wise information. Moreover, we also provide theoretical support for the RBC module and verify the generalization error bound.

Existing subgraph-aware self-supervised learning methods (Feng et al., 2019; Dash et al., 2021; 2022; Yang et al., 2022) cannot be directly applied to train our RBC since it is different from the architectures they used. Therefore, we propose a coordinated self-supervised learning strategy for RBC. For the atom branch, we employ the commonly used masked atom prediction task, i.e., predicting randomly masked atoms. For the subgraph branch, most existing self-supervised molecular learning methods, which are mainly designed for atom-based architectures, cannot fully capture subgraph-wise information and the relations among substructures. To this end, we propose a Masked Subgraph-Token Modeling (MSTM) strategy for the subgraph-wise branch. MSTM first tokenizes a given molecule into pieces and forms a subgraph-token dictionary. Compared with atom tokens, such subgraphs correspond to different functional groups, thus their semantics are more stable and consistent. MSTM decomposes each molecule into subgraph tokens, masks a portion of them, and learns the molecular representation by taking the prediction of masked token indexes in the dictionary as the self-supervised task. Although the atom-wise branch and the subgraph-wise branch aim to extract molecular features from different levels, the global graph representations for the same molecule should be consistent. To build the synergistic interaction between the two branches for joint pre-training, we perform contrastive learning to maximize the average invariance of the two branches. Experimental results show the effectiveness of our method.

Our contributions can be summarized as:

1. We propose a novel architecture for the molecular property prediction task, which consists of the atom-wise branch and the decomposition-polymerization subgraph-wise branch. Moreover, a Route-guided Bilateral Compenstation (RBC) architecture is proposed to fuse the information of two branches dynamically. We verify that the two branches play a complementary role theoretically and give the generalization error bound of our RBC.

2. We propose a cooperative node-level and graph-level self-supervised learning method to jointly train the two branches of our bilateral model. For the subgraph branch, we propose MSTM, a novel self-supervised molecular learning strategy, which uses the auto-discovered subgraphs as tokens and predicts the dictionary indexes of masked tokens. The subgraph tokens are more stable in function and have more consistent semantics. In this way, masked subgraph modeling can be performed in a principled manner. At the global graph level, we perform a contrastive learning strategy that imposes the interaction of the two branches with the consistency constraint.

3. We provide extensive empirical evaluations to show that the learned representation by our bilateral model and our self-supervised learning method has a stronger generalization ability in various functional group-related molecular property prediction tasks.

## 2 RELATED WORK

**Molecular property prediction**    The prediction of molecular properties is an important research topic in the fields of chemistry, materials science, pharmacy, biology, physics, etc (Wang & Hou,

2011). Since it is time-consuming and labor-intensive to measure properties via traditional wet experiments, many recent works focus on designing end-to-end machine learning methods to directly predict properties. These works can be divided into two categories: SMILES string-based methods (Butler et al., 2018; Dong et al., 2018) and graph-based methods (Gilmer et al., 2017; Yang et al., 2019; Lu et al., 2019; Gasteiger et al., 2020). Compared with SMILES strings, it is more natural to represent molecules as graphs and model them with Graph neural networks (GNNs). However, the training of GNNs requires a large amount of labeled molecule data and supervised-trained GNNs usually show limited generalization ability for newly synthesized molecules and new properties. In order to tackle these issues, self-supervised representation pre-training techniques are explored (Rong et al., 2020; Li et al., 2021; Stärk et al., 2022) in molecular property prediction.

**Self-supervised learning of graphs** Based on how self-supervised tasks are constructed, previous works can be classified into two categories, contrastive models and predictive models. Contrastive models (Li et al., 2024; Zhang et al., 2020; Sun et al., 2020; You et al., 2021; Sun et al., 2021; Subramonian, 2021; Xia et al., 2022; Li et al., 2022b; Zhong et al., 2024) generate different views for each graph via data augmentation and learn representations by contrasting the similarities between views of the same graph and different graphs. Predictive models (Hu et al., 2020; Rong et al., 2020; Hou et al., 2022) generally mask a part of the graph and predict the masked parts. Most existing methods focus on learning node-level or graph-level representations, with some work involving subgraph-level feature that utilizes the rich semantic information contained in the subgraphs or motifs. For instance, in (Zhang et al., 2021), the topology information of motifs is considered. In (Wu et al., 2023), a Transformer architecture is proposed to incorporate motifs and construct 3D heterogeneous molecular graphs for representation learning. Different from these works, we propose a bilateral fusion model with a novel subgraph-aware GNN branch and propose a joint node-wise and graph-wise self-supervised training strategy so that the learned representation can capture both atom-wise and subgraph-wise information.

# 3 METHODOLOGY

## 3.1 PROBLEM FORMULATION

We represent a molecule as a graph $G = (V, E)$ with node attribute vectors $\boldsymbol{x}_v$ for $v \in V$ and edge attribute vectors $\boldsymbol{e}_{uv}$ for $(u, v) \in E$, where $V$ and $E$ are the sets of atoms and bonds, respectively. We consider a binary classification problem with instance $G$ and label $y = \{0, 1\}$, where $y$ denotes whether this property is present in $G$. Given a set of training samples $\mathcal{D}_s = \{(G_i, y_i)\}_{i=1}^{N_1}$, our target is to learn a hypothesis $f$ making predictions that can well generalize to the test set. We also have a set of unlabelled support set $\mathcal{D}_u = \{(G_i)\}_{i=1}^{N_2}$, where $N_2 \gg N_1$, and apply our self-supervised learning method to get better initial representation.

## 3.2 ATOM-WISE BRANCH

Previous works extract the representation of a molecule by aggregating the embeddings of all atoms with GNNs. Similarly, our atom-wise branch applies a single GNN model with $K$ layers to map each molecule graph into an embedding. Specifically, for $G = (V, E)$, the input embedding $\boldsymbol{h}_v^0$ of the node $v \in V$ is initialized by $\boldsymbol{x}_v$, the input embedding at the $k$-th layer $\boldsymbol{e}_{uv}^k$ of the edge $(u, v) \in E$ is initialized by $\boldsymbol{e}_{uv}$, and the $K$ GNN layers iteratively update $\boldsymbol{h}_v$ by polymerizing the embeddings of neighboring nodes and edges of $\hat{v}$. In the $k$-th layer, $\boldsymbol{h}_v^{(k)}$ is updated as follows:

$$\boldsymbol{h}_v^{(k)} = \text{COMBINE}^{(k)}(\boldsymbol{h}_v^{(k-1)}, \text{AGGREGATE}^{(k)}$$
$$(\{(\boldsymbol{h}_v^{(k-1)}, \boldsymbol{h}_u^{(k-1)}, \boldsymbol{e}_{uv}^k) : u \in \mathcal{N}(v)\})) \quad (1)$$

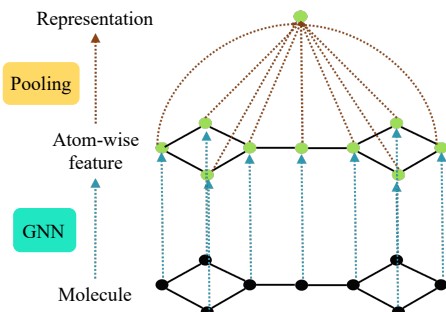

Figure 1: The atom-wise branch.

where $\boldsymbol{h}_v^{(k)}$ denotes the embedding of node $v$ at the $k$-th layer, and $\mathcal{N}(v)$ represents the neighborhood set of node $v$. After $K$ iterations of aggregation, $\boldsymbol{h}_v^{(K)}$ captures the structural information within its $K$-hop network neighborhoods. The embedding $\boldsymbol{z}_A = \text{MEAN}(\{\boldsymbol{h}_v^{(K)} | v \in V\})$ of the graph $G$ is the average of each node. Then we add a linear classifier to achieve the final prediction. Formally, we denote $f$ as the atom-wise branch and the supervised loss is $\ell_1 = \ell(f(G), y)$.

### 3.3 Subgraph-wise branch

Atoms are influenced by their surrounding contexts and the semantics of a single atom can change significantly in different environments. Functional groups, which are connected subgraphs composed of coordinated atoms, determine many molecular properties. Our proposed hierarchical Decomposition-Polymerization architecture decouples the representation learning into the subgraph embedding phase, where each molecule is decomposed into subgraphs and an embedding vector is extracted from each subgraph, and the subgraph polymerization phase, where subgraphs are modeled as nodes and their embeddings are updated by polymerizing information from neighboring subgraphs. Finally, the final representation is obtained by combining all subgraph-wise embeddings.

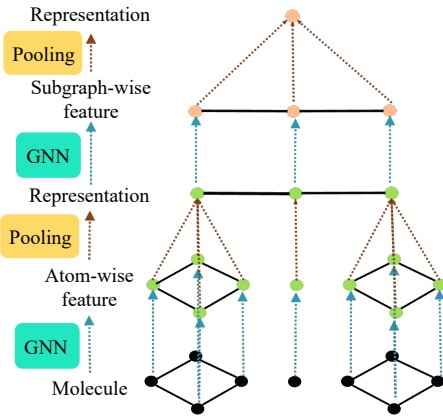

Figure 2: The subgraph-wise branch.

**Subgraph vocabulary construction** Functional groups correspond to special subgraphs, however, pre-defined subgraph vocabularies of hand-crafted functional groups may be incomplete, i.e., not all molecules can be decomposed into disjoint subgraphs in the vocabulary. There exist many decomposition algorithms such as the principle subgraph extraction strategy (Kong et al., 2022) and breaking retrosynthetically interesting chemical substructures (BRICS) (Degen et al., 2008). Generally, we denote a subgraph of the molecule $G$ by $S = (\hat{V}, \hat{E}) \subset G$, where $\hat{V}$ is a subset of $V$ and $\hat{E}$ is the subset of $E$ corresponding to $\hat{V}$. The target of principle subgraph extraction is to constitute a vocabulary of subgraphs $\mathbb{V} = \{S_{(1)}, S_{(2)}, \cdots, S_{(M)}\}$ that represents the meaningful patterns within molecules, where each unique pattern is associated with an index.

**Subgraph embedding** In this phase, we only focus on learning the embedding of each subgraph by modeling the intra-subgraph interactions. For a molecule $G = (V, E)$, we decompose it into a set of non-overlapped subgraphs $\{S_{\pi 1}, S_{\pi 2}, \cdots, S_{\pi T}\}$, where $T$ is the number of decomposed subgraphs and $\pi t$ is the corresponding index of the $t^{th}$ decomposed subgraph in the constructed vocabulary $\mathbb{V}$. For each subgraph $S_{\pi t} = (\hat{V}_{\pi t}, \hat{E}_{\pi t})$, we have $\hat{V}_{\pi t} \subset V$ and $\hat{E}_{\pi t} \subset E$. For each edge $(u, v)$ in $E$, we add it into the inter-subgraph edge set $\mathcal{E}$ if it satisfies that nodes $u$ and $v$ are in different subgraphs. Therefore, we have $V = \cup \hat{V}_{\pi t}$ and $E = \cup \hat{E}_{\pi t} \cup \mathcal{E}$.

We apply a single GNN model with $K_1$ layers to map each decomposed subgraph into an embedding. GNN depends on the graph connectivity as well as node and edge features to learn an embedding for each node $v$. We discard the inter-subgraph edge set $\mathcal{E}$, any two subgraphs are disconnected and the information will be detached among subgraphs. This is equivalent to feeding each subgraph $S_{\pi t}$ into the GNN model individually.

By feeding the molecular graph after discarding all inter-subgraph edges into the GNN model, the embeddings of all atoms in the $T$ decomposed subgraphs are updated in parallel and the embeddings of all subgraphs can be obtained by adaptive pooling. Compared with previous strategies (Hu et al., 2019; Zhang et al., 2021) that directly obtain molecular representations from the context-dependent atom-wise embeddings with all edges, our strategy first extracts subgraph-level embeddings. When a subgraph appears in different molecules, both its atom-wise embeddings and the subgraph embedding remain the same.

**Subgraph-wise polymerization**   In the previous subgraph embedding phase, we view each atom in the subgraph as a node and extract the embedding of each subgraph. In the subgraph-wise poly-merization phase, we polymerize the embeddings of neighboring subgraphs for acquiring representations of subgraphs and the final representation of the molecule $G$. Differently, we view each subgraph as a node and connect them by the set of inter-subgraph edges $\mathcal{E}$. Two subgraphs $S_{\pi t}$ and $S_{\pi l}$ are connected if there exists at least one edge $(\hat{u}, \hat{v}) \in \mathcal{E}$ where $\hat{u} \in \hat{V}_{\pi t}$ and $\hat{v} \in \hat{V}_{\pi l}$. In this way, we construct another graph whose nodes are subgraphs and employ another GNN model with $K_2$ layers to update the representation of each subgraph and extract the final representation $\boldsymbol{z}_S$. At the $k'$-th layer, the embedding $\boldsymbol{h}_{\pi t}$ for the $t$-th subgraph is updated as follows:

$$\boldsymbol{h}_{\pi t}^{(k')} = \text{COMBINE}^{(k')}(\boldsymbol{h}_{\pi t}^{(k'-1)}, \text{AGGREGATE}^{(k')}(\{\boldsymbol{h}_{\pi t}^{(k'-1)}, \boldsymbol{h}_{\pi l}^{(k'-1)}, e_{\hat{u}\hat{v}}^{k'}) : (\hat{u}, \hat{v}) \in \mathcal{E} \\ \text{AND} \quad \hat{u} \in \hat{V}_{\pi t} \quad \text{AND} \quad \hat{v} \in \hat{V}_{\pi l}\}) \tag{2}$$

The representation $\boldsymbol{z}_S = \text{MEAN}(\{\boldsymbol{h}_{\pi t}^{(K_2)} | t \in \{1, 2, \cdots, T\}\})$ has aggregated all information from different subgraphs, where $\boldsymbol{h}_{\pi t}^{(K_2)}$ denotes the subgraph feature which is fed forward after $K_2$ iterations. Formally, we denote $g$ as the subgraph-wise branch and the supervised loss is $\ell_2 = \ell(g(G), y)$.

## 3.4   Route-guided Bilateral Compensation (RBC)

The properties of some molecules are determined by their constituent atoms, while others are influenced by functional groups, and in most cases, it's a combination of both factors. When the property is more closely related to the atom, we should utilize more information from the atom-wise branch, and vice versa. The current fusion methods often directly aggregate information, which may not be sufficiently accurate (Wang et al., 2022; Fey et al., 2020). Therefore, we introduce our Route-guided Bilateral Compensation (RBC) to control the aggregation rate automatically. Formally, we define the fusion feature as shown in Eq. 3, where $\lambda = \frac{\alpha_1 + c/2}{\alpha_1 + \alpha_2 + c}$ represent the aggregation rate. We define $\alpha_1 = S(\boldsymbol{W}_1^T \boldsymbol{z}_A)$ and $\alpha_2 = S(\boldsymbol{W}_2^T \boldsymbol{z}_S)$ and $c$ is a scaling factor, where $S(\cdot)$ is the sigmoid function, and $\boldsymbol{W}_1$ and $\boldsymbol{W}_2$ are two learnable parameters.

$$\tilde{\boldsymbol{z}} = \lambda \cdot \boldsymbol{z}_A + (1 - \lambda) \cdot \boldsymbol{z}_S \tag{3}$$

In our approach, the ultimate output feature, denoted as $\tilde{\boldsymbol{z}}$, combines the invariant features from both branches, incorporating the weighted contributions from both the atom-wise and subgraph-wise branches. Typically, when a property exhibits a stronger association with subgraphs, the value of $\alpha_2$ surpasses that of $\alpha_1$, and conversely for properties leaning towards atoms. Our route mechanism adeptly automates the feature selection process, effectively enhancing overall performance. Finally, we add a linear classifier to realize the classification task.

## 3.5   Theoetrical analysis

**The complementarity between atom-wise branch and subgraph-wise branch**   Intuitively, atom-wise information dictates certain properties, while subgraph-wise information governs others. Merging these two types of information can enhance property prediction in a complementary manner (Yang et al., 2022). However, this intuition alone does not fully explain the workings of our architecture. In this section, we analyze our method from the perspective of GNN expressiveness and provide a plausible explanation.

**Definition 1** *Given two mapping functions $m_1$ and $m_2$, if there exists two non-isomorphic molecular graphs $G_1$ and $G_2$ such that $m_1(G_1) = m_1(G_2)$ whereas $m_2(G_1) \neq m_2(G_2)$. We denote $m_2$ is more expressive than $m_1$.*

**Theorem 1** *For atom-wise branch $f$ and subgraph-wise branch $g$, $g$ is more expressive than $f$.*

The Theorem 1 highlights that the subgraph-wise branch can distinguish a greater number of non-isomorphic graphs compared to the atom-wise branch. However, expressive ability does not necessarily equate to discriminative ability, which is critical for making property predictions. To address this, the following corollary gives a further explanation of how the two branches work.

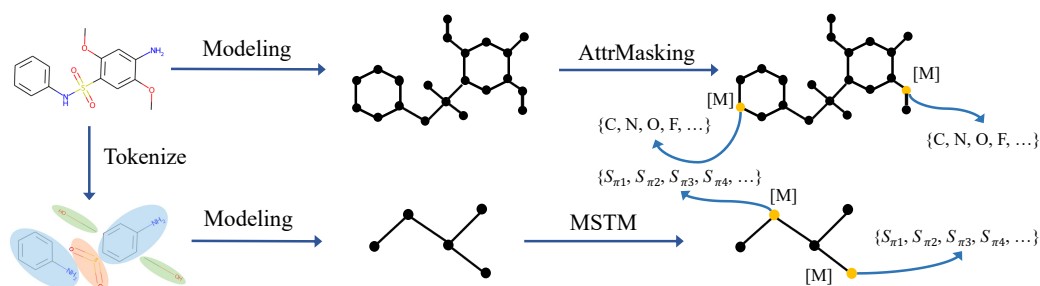

Figure 3: The differences between AttrMasking (Hu et al., 2019) and our MSTM.

**Corollary 1** *Given two molecular samples $(G_1, y_1)$ and $(G_2, y_2)$. If $y_1 = y_2$, then the atom-wise branch is more important to make a judgment that they have a similar property.*

**Corollary 2** *Given two molecular samples $(G_1, y_1)$ and $(G_2, y_2)$. If $y_1 \neq y_2$, then the subgraph-wise branch is more important to make a judgment that they have different properties.*

Corollary 1 and Corollary 2 illustrate the compensation of two branches. Considering two molecular graphs with identical properties, the optimal pair of representations extracted by the mapping function will be as similar as possible within the feature space. However, a mapping function with greater expressive power might overly differentiate between these graphs, potentially impairing property prediction. Conversely, when dealing with two molecular graphs exhibiting distinct properties, a more expressive mapping function is required to effectively capture their differences, placing greater emphasis on the subgraph-wise branch. Detailed proofs are provided in the appendix.

**RBC lead to better generalization**   Previous analyses indicate that the atom-wise and subgraph-wise branches provide complementary information. Integrating them is expected to yield superior performance compared to relying on a single branch alone. Inspired by (Zhang et al., 2023), we give the generalization error bound $\epsilon(f \circ g)$ for the methods following the fusion strategy in Eq. 3, where $f$ and $g$ denote the atom-wise and subgraph-wise branch.

**Assumption 1** *Let $Z$ and $Y$ be two random variables from the latent feature space and output space, associated with our model, respectively. We assume that the mutual information between $Z$ and $Y$, denote as $\mathcal{I}(Z, Y)$, satisfies $\mathcal{I}(Z, Y) \gg \Delta$, where $\Delta$ is a large constant, indicating a strong dependence between $Z$ and $Y$.*

**Theorem 2** *Let $\hat{E}(f)$ and $\hat{E}(g)$ denote the empirical errors of model $f$ and $g$ on the training data $\mathcal{D}_s$, respectively. $\mathcal{R}$ is the Rademacher complexity and $Cov(\cdot, \cdot)$ is the covariance between score and loss. We denote $\epsilon(f \circ g)$ as the generalized error bound of our RBC. With probability at least $1 - \delta$ ($0 < \delta < 1$), it is hold:*

$$\epsilon(f \circ g) < \mathbb{E}(\lambda)\hat{E}(f) + \mathbb{E}(1 - \lambda)\hat{E}(g) + \mathbb{E}(\lambda)\mathcal{R}(f) + \mathbb{E}(1 - \lambda)\mathcal{R}(g) +$$

$$Cov(\lambda, \ell_1) - Cov(\lambda, \ell_2) + 2\sqrt{\frac{ln(1/\delta)}{2N_1}} \tag{4}$$

The Theorem 2 presents that the error bound is constituted by empirical errors, model complexities, and covariances between fusion weights and branch losses. Among different integration strategies, if we fix the expectations of fusion weight, then the covariances term will influence the generalization bound mainly. In the experiments section, we will experimentally verify that our RBC can achieve a lower value of covariances and lead to better generalization ability.

### 3.6   SELF-SUPERVISED LEARNING

**Node-level self-supervised learning**   Many recent works show that self-supervised learning can learn generalizable representations from a large number of unlabelled molecules. Since the atom-

wise branch and subgraph-wise branch are decoupled, we can apply the existing atom-wise self-supervised learning method to the atom-wise branch of RBC such as attrMasking (Hu et al., 2019).

For the subgraph-wise branch, we propose the Masked Subgraph-Token Modeling (MSTM) strategy, which randomly masks some percentage of subgraphs and then predicts the corresponding subgraph tokens. As shown in Fig.3, a training molecule $G$ is decomposed into $T$ subgraphs $\{S_{\pi 1}, S_{\pi 2}, \cdots, S_{\pi T}\}$. The subgraphs are tokenized to tokens $\{\pi 1, \pi 2, \cdots \pi T\}$, respectively. Similar to BEiT (Bao et al., 2021), we randomly mask a number of $M$ subgraphs and replace them with a learnable embedding. Therefore, we construct a corrupted graph $\tilde{G}$ and feed it into our hierarchical decomposition-polymerization GNN architecture to acquire polymerized representations of all subgraphs. For each masked subgraph $\tilde{S}_{\pi t}$, we bring an MSTM classifier $p(\cdot|h_{\pi t})$ with weight $\boldsymbol{W}_p$ and bias $\boldsymbol{b}_p$ to predict the ground truth token $\pi t$. The pre-training objective of MSTM is to minimize the negative log-likelihood of the correct tokens given the corrupted graphs.

$$\mathcal{L}_{\text{MSTM}} = \frac{1}{N_2} \sum_{\tilde{G} \in \mathcal{D}_u} -\mathbb{E}\left[\sum_t \log p_{\text{MSTM}}(\pi t | g(\tilde{G}))\right] \tag{5}$$

where $p_{\text{MSTM}}(\pi t | \tilde{G}) = \text{Softmax}(\boldsymbol{W}_p \tilde{\boldsymbol{h}}_{\pi t}^{(K_2)} + \boldsymbol{b}_p)$. Different from previous strategies, which randomly mask atoms or edges to predict the attributes, our method randomly masks some subgraphs and predicts their indices in the vocabulary $\mathbb{V}$ with the proposed decomposition-polymerization architecture. Actually, our prediction task is more difficult since it operates on subgraphs and the size of $\mathbb{V}$ is larger than the size of atom types. As a result, the learned substructure-aware representation captures high-level semantics of substructures and their interactions and can be better generalized to the combinations of known subgraphs under different scaffolds.

**Graph-level self-supervised learning** Node-level pre-training alone is insufficient for obtaining features that can be generalized (Xia et al., 2023). Therefore, we propose graph-level self-supervised learning, as illustrated in Eq. 6, where $\mathcal{B}^-$ represents negative samples for the anchor sample $G_i$. These negative samples are comprised of the remaining samples within the same batch. We define $\boldsymbol{v}_i = (\boldsymbol{z}_{A_i} + \boldsymbol{z}_{S_i})/2$, and $(\boldsymbol{v}_i, \boldsymbol{v}_i^{'})$ constitutes a pair of augmentation graphs derived from $G_i$. In the atom-wise branch, we randomly remove some atoms, and in the subgraph-wise branch, we randomly remove one subgraph to implement augmentation.

$$\mathcal{L}_{cl} = \frac{1}{N_2} \sum_{G_i \in \mathcal{D}_u} -\log \frac{\exp\left(\boldsymbol{v}_i \cdot \boldsymbol{v}_i^{'}\right)}{\exp\left(\boldsymbol{v}_i \cdot \boldsymbol{v}_i^{'}\right) + \sum_{G_j \in \mathcal{B}^-} \exp\left(\boldsymbol{v}_i \cdot \boldsymbol{v}_j\right)} \tag{6}$$

Our method is different from GraphCL (You et al., 2020) and Mole-Bert Xia et al. (2023), which apply graph-level augmentation on the atom-wise branch only. Our method maximizes the feature invariance along these two branches and improves the generalization for downstream tasks. In addition, graph-level self-supervised learning makes the two branches interact which can utilize the superiority of our bilateral architecture.

## 4 EXPERIMENTS

### 4.1 DATASETS AND EXPERIMENTAL SETUP

**Datasets and Dataset Splittings** We use the ZINC250K dataset (Sterling & Irwin, 2015) for self-supervised pre-training, which is constituted of $250k$ molecules up to 38 atoms. As for downstream molecular property prediction tasks, we test our method on 8 classification tasks and 3 regression tasks from MoleculeNet (Wu et al., 2018). For classification tasks, we follow the *scaffold-splitting* (Ramsundar et al., 2019), where molecules are split according to their scaffolds (molecular substructures). The proportion of the number of molecules in the training, validation, and test sets is $80\% : 10\% : 10\%$. Following (Li et al., 2022a), we apply random scaffold splitting to regression tasks, where the proportion of the number of molecules in the training, validation, and test sets is also $80\% : 10\% : 10\%$. Following (Zhang et al., 2021; Liu et al., 2021), we performed 10 replicates on each dataset to obtain the mean and standard deviation.

Table 1: Test ROC-AUC performance of different methods on molecular property classification tasks. AVG represents the average results overall benchmarks. We highlight the best and second-best results with * and *. We report the mean and standard results.

| Methods | BACE | BBBP | ClinTox | HIV | MUV | SIDER | Tox21 | ToxCast | Avg |
|---|---|---|---|---|---|---|---|---|---|
| Infomax | 75.9(1.6) | 68.8(0.8) | 69.9(3.0) | 76.0(0.7) | 75.3(2.5) | 58.4(0.8) | 75.3(0.5) | 62.7(0.4) | 70.3 |
| AttrMasking | 79.3(1.6) | 64.3(2.8) | 71.8(4.1) | 77.2(1.1) | 74.7(1.4) | 61.0(0.7) | 76.7(0.4) | 64.2(0.5) | 71.1 |
| GraphCL | 75.4(1.4) | 69.7(0.7) | 76.0(2.7) | 78.5(1.2) | 69.8(2.7) | 60.5(0.9) | 73.9(0.7) | 62.4(0.6) | 70.8 |
| AD-GCL | 78.5(0.8) | 70.0(1.1) | 79.8(3.5) | 78.3(1.0) | 72.3(1.6) | 63.3(0.8) | 76.5(0.8) | 63.1(0.7) | 72.7 |
| MGSSL | 79.1(0.9) | 69.7(0.9) | 80.7(2.1) | **78.8(1.2)** | **78.7(1.5)** | 61.8(0.8) | 76.5(0.3) | 64.1(0.7) | 73.7 |
| GraphLoG | **83.5(1.2)** | 72.5(0.8) | 76.7(3.3) | 77.8(0.8) | 76.0(1.1) | 61.2(1.1) | 75.7(0.5) | 63.5(0.7) | 73.4 |
| GraphMVP | 81.2(0.9) | 72.4(1.6) | 77.5(4.2) | 77.0(1.2) | 75.0(1.0) | **63.9(1.2)** | 74.4(0.2) | 63.1(0.4) | 73.1 |
| GraphMAE | 83.1(0.9) | 72.0(0.6) | **82.3(1.2)** | 77.2(1.0) | 76.3(2.4) | 60.3(1.1) | 75.5(0.6) | 64.1(0.3) | 73.8 |
| Mole-Bert | 80.8(1.4) | 71.9(1.6) | 78.9(3.0) | 78.2(0.8) | 78.6(1.8) | 62.8(1.1) | **76.8(0.5)** | 64.3(0.2) | 74.0 |
| RBC | 81.2(1.5) | **74.2(0.3)** | 80.9(1.9) | 78.6(1.0) | 77.5(1.2) | 63.4(0.9) | 76.8(1.0) | **65.3(0.3)** | 74.7 |

Table 2: Test RMSE performance of different methods on the regression datasets.

| Methods | Regression dataset | | | | | |
|---|---|---|---|---|---|---|
| | fine-tuning | | | linear probing | | |
| | FreeSolv | ESOL | Lipo | FreeSolv | ESOL | Lipo |
| Infomax | 3.416(0.928) | 1.096(0.116) | 0.799(0.047) | 4.119(0.974) | 1.462(0.076) | 0.978(0.076) |
| EdgePred | 3.076(0.585) | 1.228(0.073) | 0.719(0.013) | 3.849(0.950) | 2.272(0.213) | 1.030(0.024) |
| Masking | 3.040(0.334) | 1.326(0.115) | 0.724(0.012) | 3.646(0.947) | 2.100(0.040) | 1.063(0.028) |
| ContextPred | 2.890(1.077) | 1.077(0.029) | 0.722(0.034) | 3.141(0.905) | 1.349(0.069) | 0.969(0.076) |
| GraphLog | 2.961(0.847) | 1.249(0.010) | 0.780(0.020) | 4.174(1.077) | 2.335(0.073) | 1.104(0.024) |
| GraphCL | 3.149(0.273) | 1.540(0.086) | 0.777(0.034) | 4.014(1.361) | 1.835(0.111) | 0.945(0.024) |
| GraphMVP | 2.874(0.756) | 1.355(0.038) | 0.712(0.025) | **2.532(0.247)** | 1.937(0.147) | 0.990(0.024) |
| RBC | **2.793(0.689)** | **0.922(0.102)** | **0.533(0.012)** | 3.010(0.734) | **1.268(0.204)** | **0.810(0.053)** |

**Baselines** For classification tasks, we comprehensively evaluated our method against different self-supervised learning methods on molecular graphs, including Infomax (Veličković et al., 2018), AttrMasking (Hu et al., 2019), ContextPred (Hu et al., 2019), GraphCL (You et al., 2020), AD-GCL (Suresh et al., 2021), MGSSL (Zhang et al., 2021), GraphLog (Xu et al., 2021), Graph-MVP (Liu et al., 2021), GraphMAE (Hou et al., 2022), and Mole-Bert Xia et al. (2023). For regression tasks, we compare our method with Infomax (Veličković et al., 2018), EdgePred (Hamilton et al., 2017), AttrMasking (Hu et al., 2019), ContextPred (Hu et al., 2019), GraphLog (Xu et al., 2021), GraphCL (You et al., 2020),and GraphMVP (Liu et al., 2021). Among them, 3DInfomax exploits the three-dimensional structure information of molecules, while other methods also do not use knowledge or information other than molecular graphs.

## 4.2 RESULTS AND ANALYSIS

**Classification** Tab. 1 presents the results of fine-tuning compared with the baselines on classification tasks. "RBC" denotes the results of our method after self-supervised per-training. From the results, we observe that the overall performance of our method is significantly better than all baseline methods on most datasets. Among them, AttrMasking and GraphMAE also use masking strategies that operate on atoms and bonds in molecular graphs. Compared with AttrMasking, our method achieves a significant performance improvement of 10.0%, 9.1%, and 2.4% on BBBP, ClinTox, and MUV datasets respectively, with an average improvement of 3.2% on all datasets. Compared with GraphMAE, our method also achieved a universal improvement. Compared with contrastive learning models, our method achieves a significant improvement with an average improvement of 4.0% compared with Infomax, 3.5% compared with GraphCL, 1.6% compared with AD-GCL, and 0.9% compared with GraphLoG. For GraphMVP which combines contrastive and generative methods, our method also has an average improvement of 1.2%.

Table 3: Test ROC-AUC performance of different methods on molecular property classification tasks with different tokenization algorithms and model configurations.

| Methods | BACE | BBBP | ClinTox | HIV | MUV | SIDER | Tox21 | ToxCast | Avg |
|---|---|---|---|---|---|---|---|---|---|
| Atom-wise | 71.6(4.5) | 68.7(2.5) | 57.5(3.8) | 75.6(1.4) | 73.2(2.5) | 57.4(1.1) | 74.1(1.4) | 62.4(1.0) | 67.6 |
| The principle subgraph, $\lvert\mathbb{V}\rvert = 100$, $K_1 = 2$, $K_2 = 3$ | | | | | | | | | |
| Subgraph-wise | 64.4(5.2) | 69.4(3.0) | 59.2(5.2) | 71.7(1.5) | 68.3(1.6) | 59.1(0.9) | 72.6(0.8) | 61.3(0.7) | 65.8 |
| RBC | 72.1(2.8) | 72.4(2.3) | 57.0(4.8) | 74.7(1.9) | 71.4(1.8) | 59.6(1.6) | 76.1(0.8) | 63.8(0.6) | 68.4 |
| The principle subgraph, $\lvert\mathbb{V}\rvert = 100$, $K_1 = 3$, $K_2 = 2$ | | | | | | | | | |
| Subgraph-wise | 63.1(7.1) | 68.7(2.7) | 55.8(6.3) | 71.7(1.7) | 68.3(4.9) | 58.7(1.5) | 72.9(0.9) | 61.5(0.9) | 65.1 |
| RBC | 73.4(3.3) | 70.7(2.9) | 59.5(4.2) | 74.8(1.1) | 71.8(2.8) | 59.2(1.7) | 75.8(0.7) | 64.4(0.6) | 68.7 |
| The principle subgraph, $\lvert\mathbb{V}\rvert = 300$, $K_1 = 2$, $K_2 = 3$ | | | | | | | | | |
| Subgraph-wise | 66.2(4.5) | 63.7(3.2) | 59.0(8.5) | 74.2(1.6) | 68.9(1.9) | 61.6(1.8) | 73.3(0.9) | 60.5(0.5) | 65.9 |
| RBC | 69.3(4.5) | 67.3(4.1) | 62.7(5.2) | 76.2(2.1) | 72.2(2.9) | 60.1(2.1) | 76.0(0.8) | 63.7(0.7) | 68.4 |
| The principle subgraph, $\lvert\mathbb{V}\rvert = 300$, $K_1 = 3$, $K_2 = 2$ | | | | | | | | | |
| Subgraph-wise | 67.3(2.0) | 66.5(3.3) | 54.7(5.7) | 73.8(2.1) | 69.7(2.5) | 60.8(2.5) | 73.7(0.7) | 60.8(0.7) | 65.9 |
| RBC | 74.8(2.8) | 69.4(2.9) | 57.0(3.9) | 77.1(0.9) | 72.5(2.2) | 60.5(1.6) | 75.9(0.7) | 63.8(0.7) | 68.9 |
| BRICS, $K_1 = 2$, $K_2 = 3$ | | | | | | | | | |
| Subgraph-wise | 71.4(3.9) | 66.1(3.5) | 51.8(3.7) | 75.2(1.8) | 70.0(2.0) | 56.0(1.5) | 74.0(0.8) | 64.2(1.1) | 66.1 |
| RBC | 72.4(4.3) | 69.6(1.8) | 60.8(6.7) | 75.9(1.5) | 73.3(2.4) | 58.1(1.3) | 75.8(0.6) | 65.5(0.7) | 68.9 |
| BRICS, $K_1 = 3$, $K_2 = 2$ | | | | | | | | | |
| Subgraph-wise | 73.6(3.7) | 67.0(1.9) | 53.0(5.3) | 74.0(1.7) | 70.5(1.9) | 55.8(1.7) | 74.4(1.0) | 65.0(0.4) | 66.7 |
| RBC | 72.4(4.6) | 69.9(1.8) | 53.5(11.5) | 77.0(1.0) | 73.9(1.9) | 55.7(1.5) | 75.7(0.8) | 64.9(1.1) | 67.9 |

Table 4: Ablation study on different components of our dual branch self-supervised learning method.

| Methods | BACE | BBBP | ClinTox | HIV | MUV | SIDER | Tox21 | ToxCast | Avg |
|---|---|---|---|---|---|---|---|---|---|
| Node-level | 78.8(1.9) | 72.8(1.4) | 74.2(2.8) | 77.1(0.6) | 73.7(1.5) | 61.3(0.9) | 75.9(0.5) | 65.6(0.3) | 72.4 |
| Graph-level | 73.9(1.4) | 69.5(2.3) | 61.8(3.2) | 75.6(1.0) | 73.1(1.9) | 59.0(0.8) | 74.6(0.3) | 63.1(0.5) | 68.8 |
| Node+Graph | 81.2(1.5) | 74.2(0.3) | 80.9(1.9) | 78.6(1.0) | 77.5(1.2) | 63.4(0.9) | 76.8(1.0) | 65.3(0.3) | 74.7 |

**Regression** In Tab. 2, we report evaluation results in regression tasks under the fine-tuning and linear probing protocols for molecular property prediction. Other methods are pre-trained on the large-scale dataset ChEMBL29 (Gaulton et al., 2012) containing 2 million molecules, which is 10 times the size of the dataset for pre-training our method. The comparison results show that our method outperforms other methods and achieves the best performance in five out of six tasks, despite being pre-trained only on a small-scale dataset. This indicates that our method can better learn transferable information about atoms and subgraphs from fewer molecules with higher data-utilization efficiency.

**RBC can achieve better generalization** In Tab. 3, we compare the performance of our atom-level, subgraph-level, and integrated RBC model on different classification tasks. From the experimental results, it can be seen that the atom-level branch performs better than the subgraph-level branch on some datasets, such as BACE, ToxCast, and Tox21, while the subgraph-level branch outperforms on others, such as SIDER and BBBP. This is because the influencing factors of different classification tasks are different, some focus on functional groups, while some focus on the interactions between atoms and chemical bonds. However, no matter how the parameters of the model change, our RBC always achieves better results than the two separate branches on all datasets since it adaptatively integrates the strengths of both. These results demonstrate that our RBC has better generalization ability.

**The effectiveness of our self-supervised learning in the pre-training stage** From Tab. 1 and Tab. 2, it is evident that our self-supervised learning during the pre-training stage yields superior results in most tasks. Our self-supervised learning approach comprises node-level and graph-level

Table 6: Ablation study on the Route-guided mechanism.

| Methods | BACE | BBBP | ClinTox | HIV | MUV | SIDER | Tox21 | ToxCast | Avg |
|---|---|---|---|---|---|---|---|---|---|
| RBC (w/o route) | 77.1(3.2) | 73.8(0.8) | 80.4(3.0) | 77.2(0.8) | 77.6(0.5) | 62.8(0.6) | 75.9(0.4) | 65.2(0.3) | 73.8 |

learning components, and we conduct an independent analysis of their effectiveness. The experimental results presented in Tab. 4 indicate that joint pre-training of the two branches leveraging two self-supervised learning methods is more effective than pre-training separately (i.e., solely applying node-level or graph-level self-supervised components). To elaborate, combining both self-supervised learning components results in a 1.9% improvement compared to using node-level mask reconstruction alone and a 5.5% improvement compared to using graph-level contrastive learning alone. These findings underscore the significance of combining these two self-supervised learning components and facilitating interaction between the two branches.

**Effectiveness of Route-guided mechanism** There exist many fusion mechanisms such as fusing the features from atom-wise and subgraph-wise branches directly, i.e. we can set $\lambda = 0.5$ for each molecule. However, such a method leads to a higher value generalization error bound compared with our RBC. To theoretically prove that, we report the covariance between $\lambda$ and $l_1$, and the expectations of our fusion weight on

Table 5: Covariance between $\alpha$ and $\ell_1$ on different datasets.

| | BACE | BBBP | ClinTox |
|---|---|---|---|
| Covariance | -0.52 | -0.13 | -0.80 |

the BACE dataset during the training phase, as shown in Tab. 5. In addition, the experimental performance shown in Tab. 6 can also verify our point.

**Visualization of our Route-guided Module** We provide visual representations of selected instances along with their corresponding score values. As shown in Fig. 4, molecules with a greater number of atoms tend to allocate more attention to the subgraph-wise branch, while those with fewer atoms prioritize the atom-wise branch. This preference arises due to the increased complexity of message passing as the number of atoms grows. Within the subgraph-wise branch, we reestablish connections between different subgraphs and enhance interactions among them. Consequently, our subgraph-wise branch tends to provide more benefits to molecules with a higher number of atoms.

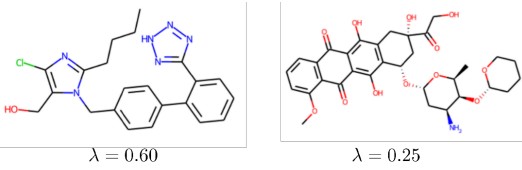

$\lambda = 0.60$      $\lambda = 0.25$

Figure 4: The fusion weights of two different molecules. The judgment of the left molecule relies on the atom-wise branch while the left relies on the subgraph-wise branch.

## 5 CONCLUSION

In this paper, we acknowledge that neither atoms nor subgraphs solely determine molecular properties and introduce a novel approach termed Route-Guided Bilateral Compensation (RBC). The RBC model consists of two branches: one dedicated to modeling atom-wise information and the other focused on subgraph-wise information. Theoretically, we prove that the subgraph-wise branch is more expressive than the atom-wise branch and integration can achieve lower generalization error bound. Furthermore, recognizing that molecular properties are influenced differently by atoms and subgraphs, we propose a route-guided mechanism for the automatic fusion of features from these two branches. To enhance generalization, we introduce a node-level self-supervised learning method called MSTM, specifically designed for the less-explored subgraph-wise branch. Additionally, we implement a graph-level self-supervised learning strategy aimed at maximizing the average invariance across the two branches. Experimental results demonstrate the effectiveness of our approach.

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

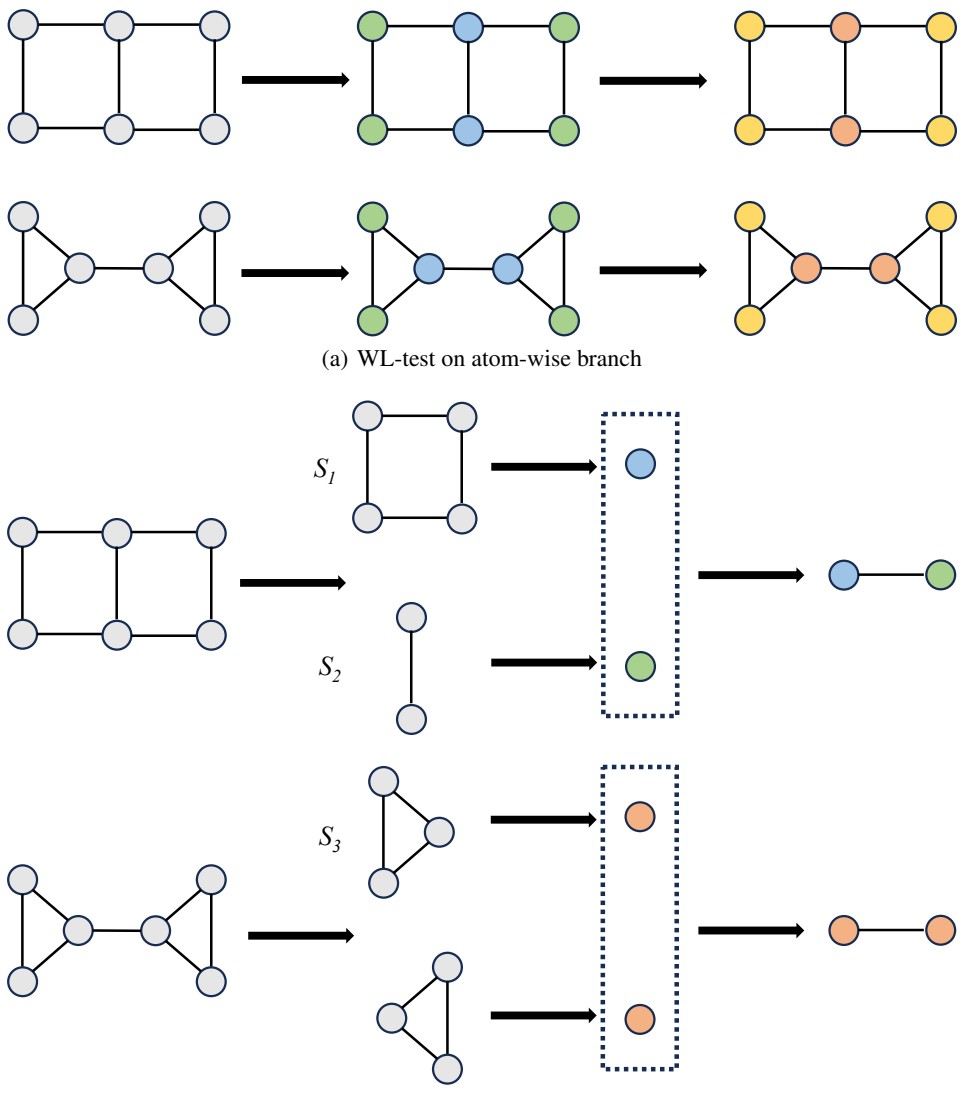

(a) WL-test on atom-wise branch

(b) WL-test on Subgraph-wise branch

Figure 5: Given two non-isomorphic graphs, the atom-wise branch fail to distinguish them while subgraph-wise does.

## A  PROOF OF THE THEORETICAL ANALYSIS

### A.1  PROOF OF THE THEOREM 1

We use the Weisfeiler & Leman Test (WL-test) to prove the subgraph-wise branch is more expressive than the atom-wise branch. The algorithm works by iteratively refining the coloring of the vertices in each graph. Initially, all vertices are assigned the same color. In each iteration, the color of a vertex is updated based on the colors of its neighbors, until the coloring stabilizes. If two graphs have the same color distribution at the end of the process, they are considered to be isomorphic under this test. However, the WL test is not complete for all graphs—it can fail to distinguish some non-isomorphic graphs, although it performs well in many practical cases, particularly for small graphs. Formally, we give the color update for color $c$ of node $v$ in iteration $t$, where the initialization value $c_v^0 = \mathrm{HASH}(\boldsymbol{x}_v)$.

$$c_v^t = \mathrm{HASH}(c_v^{t-1}, \{\{c_w^{t-1} \mid w \in \mathcal{N}(v)\}\}) \tag{7}$$

*proof.* Considering these two graphs in Fig. 5, if we extract the features by the atom-wise model, the WL test fails to distinguish these non-isomorphic graphs, since the final refinement graphs have the same hash values. Differently, if we use the subgraph-wise branch to extract the features, we can successfully distinguish them. This example demonstrates the subgraph-wise branch is more expressive than the atom-wise branch.

## A.2 PROOF OF THE THEOREM 2

*proof.* Given a pair of a sample $(G, y)$, the calculated loss of the atom-wise branch and subgraph-wise branch is $\ell_1 = \ell(f(G), y)$, $\ell_2 = \ell(g(G), y)$, respectively. Under Assumption. 1, the output feature is strongly correlated with the output logit, we can get:

$$\ell(f \circ g(G), y) = \ell(\lambda f(G, y) + (1 - \lambda)g(G, y)) \tag{8}$$

Considering $\ell$ is a convex function, and following Jensen's inequality, we have:

$$\begin{aligned} \ell(f \circ g(G), y) &= \ell(\lambda f(G, y) + (1 - \lambda)g(G, y)) \\ &\leq \lambda \ell(f(G), y) + (1 - \lambda)\ell(g(G, y) \end{aligned} \tag{9}$$

For all the molecules in the $\mathcal{D}_t$, we get:

$$\begin{aligned} \frac{1}{N_1} \sum_{i=1}^{N_1} \ell(f \circ g(G_i), y_i) &\leq \frac{1}{N_1} \sum_{i=1}^{N_1} \Big( \lambda_i \ell(f(G_i), y_i) + (1 - \lambda_i)\ell(g(G_i), y_i) \Big) \\ &= \mathbb{E}(\lambda)\mathbb{E}(\ell_1) + \mathbb{E}(1 - \lambda)\mathbb{E}(\ell_2) + Cov(\ell_1, \lambda) + Cov(\ell_2, 1 - \lambda) \end{aligned} \tag{10}$$

where $Cov$ denotes the covariance between fusion weight and loss. For simplicity, we use $\ell_1$ and $\ell_2$ to substitute the loss of atom-wise branch and subgraph-wise branch, respectively. Then, we use the Rademacher complexity measure for model complexity. Following (Zhang et al., 2023) to quantify the generalization error of unimodal models, we can get that for any hypothesis of $f$ and $g$, with probability at least $1 - \delta$ ($0 < \delta < 1$), we can get:

$$\mathbb{E}(\ell_1) \leq \hat{E}(f) + \mathfrak{R}_m(f) + \sqrt{\frac{ln(1/\delta)}{N_1}} \tag{11}$$

$$\mathbb{E}(\ell_2) \leq \hat{E}(g) + \mathfrak{R}_m(g) + \sqrt{\frac{ln(1/\delta)}{N_1}} \tag{12}$$

Finally, we get the final generalized error bound $\epsilon(f \circ g)$:

$$\begin{aligned} \epsilon(f \circ g) <& \mathbb{E}(\lambda)\hat{E}(f) + \mathbb{E}(1 - \lambda)\hat{E}(g) + \mathbb{E}(\lambda)\mathfrak{R}_m(f) + \mathbb{E}(1 - \lambda)\mathfrak{R}_m(g) + \\ & Cov(\lambda, \ell_1) - Cov(\lambda, \ell_2) + 2\sqrt{\frac{ln(1/\delta)}{2N_1}} \end{aligned} \tag{13}$$

# B TOKENIZATION ALGORITHM

We do experiments with different tokenization algorithms and we roughly introduce these methods in this section.

## B.1 INTRODUCTION OF BRICS ALGORITHM

The BRICS algorithm is one of the molecular fragmentation methods, which is an improved and optimized algorithm based on the RECAP algorithm. The BRICS algorithm introduces a better set of fragmentation rules and a set of recombinant motifs rules to form a fragmentation space. Specifically, the BRICS algorithm obtains active building blocks by segmenting active molecules. It is known that some common chemical reactions form bonds, so when segmenting molecules, BRICS segments these bonds. The algorithm splits into 16 pre-defined bonds. These 16 pre-defined bonds ensure that the split fragments are suitable for combination and applicable to combinatorial

chemistry, and these 16 pre-defined bonds are given in the form of fragment structures. When segmenting molecules, all breakable bonds are cut off at the same time to avoid redundant fragments, and if the fragments after cleavage only contain small functional groups (such as hydrogen, methyl, ethyl, propyl, and butyl), the fragments will not be cleaved again to avoid generating useless small fragments. In the process of splitting, the algorithm preserves the cyclic structure. After each bond is broken, two breakpoints are formed. RECAP algorithm directly annotates 'isotope labels' at the divided breakpoints, that is, the ids of breakable bonds, but BRICS divides firstly annotates "isotope labels" at the breakpoints, and then replaces these isotope labels with link atoms. For the RECAP algorithm, the ids of breakable bonds corresponding to the isotope labels annotated at the two breakpoints are the same, but for the BRICS algorithm, they are different, which also proves that the BRICS algorithm takes the chemical environment and surrounding substructures of each broken bond into account, and the partition effect is better. Finally, the molecules decomposed by the BRICS algorithm are a list composed of a one-step partitioned non-redundant fragment string.

### B.2 THE PRINCIPLE SUBGRAPH EXTRACTION

Principal Subgraph Extraction is a molecular fragmentation technique proposed in (Kong et al., 2022), and it consists of three primary steps: initialization, merging, and updating. The process begins by predefining the number of subgraphs to be extracted, denoted as $N$.

In the initialization step, each unique atom in the molecule is considered as an individual subgraph. Then, during the merging phase, two adjacent fragments are combined in each iteration to form a new set of subgraphs. In this context, adjacent fragments are defined as fragments containing at least one first-order neighboring node. After merging, the subgraph with the highest frequency is selected as the new subgraph for this iteration. This merging and selection process is repeated until the total number of subgraphs equals the preset value $N$.

By systematically refining subgraph selection, this method enables the extraction of meaningful and frequent substructures, facilitating more efficient molecular representation for downstream tasks.

## C  MODEL CONFIGURATION AND IMPLEMENTED DETAILS

**Supervised learning setting**  Our method involves two branches and our final loss function is shown as follows.

$$\mathcal{L}_{atom} = \frac{1}{N_1} \sum_{(G, \boldsymbol{y}) \in \mathcal{D}_s} \ell\Big(f(G), y\Big) \tag{14}$$

$$\mathcal{L}_{subgraph} = \frac{1}{N_1} \sum_{(G, \boldsymbol{y}) \in \mathcal{D}_s} \ell\Big(g(G), y\Big) \tag{15}$$

$$\mathcal{L}_{SL} = \mathcal{L}_{atom} + \mathcal{L}_{subgraph} + \mathcal{L}_{fusion} \tag{16}$$

The reason we retain $\mathcal{L}_{atom}$ and $\mathcal{L}_{subgraph}$ is that we would like to preserve the discriminative ability for the features of atom-wise and subgraph-wise independently while keeping the fusion feature discriminative at the same time. In the inference phase, we only use the fusion branch to get the prediction score.

**Self-supervised learning setting**  Our method involves node-level self-supervised learning and graph-level self-supervised learning. The final loss function is as follows.

$$\mathcal{L}_{SSL} = \mathcal{L}_{MSTM} + \mathcal{L}_{AttrMasking} + \mu \mathcal{L}_{cl} \tag{17}$$

We denote $\mathcal{L}_{AttrMasking}$ as the Attribute Masking method for the atom-wise branch. We set $\mu = 0.1$ in our experiments since we give more importance to mask reconstruction to learn the representation of two branches and contrastive learning aims to interact the two branches.

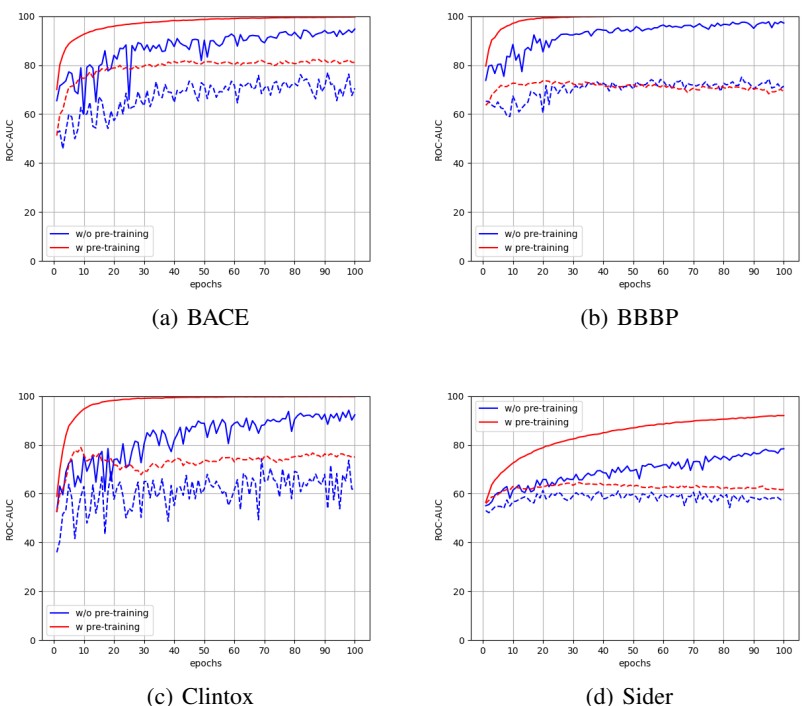

(a) BACE             (b) BBBP

(c) Clintox           (d) Sider

Figure 6: Training and testing curves. The solid lines denote training curves and the dashed lines denote testing curves. Our method shows better convergence and generalization.

**Implemented details** To validate the effectiveness of our Route-guided Bilateral Compensation (RBC) approach, we conduct experiments using different molecular fragmentation methods, such as BRICS (Degen et al., 2008) and the principle subgraph (Kong et al., 2022). Additionally, we evaluate the impact of varying hyper-parameters $K_1$ and $K_2$. In the self-supervised learning phase, we use the principle subgraph with a vocabulary size of $|\mathbb{V}| = 100$. For the subgraph embedding module, we utilize a $K_1 = 2$ layer GIN (Leskovec & Jegelka, 2019), while the subgraph-wise polymerization module employs a $K_2 = 3$ layer GIN.

In downstream tasks, including both classification and regression, we primarily follow the methodologies outlined in previous works, such as (Hu et al., 2019) for classification and (Li et al., 2022a) for regression. For RBC, we set $c = 1$ for all the datasets. During pre-training, we adopt the Adam optimizer (Kingma & Ba, 2014) with a learning rate of $1 \times 10^{-3}$, as recommended by (Zhang et al., 2021), and set the batch size to 32. This experimental setup allows for a comprehensive assessment of RBC's performance across different configurations and molecular representations.

## D VISUALIZATION CURVES

We also provide a visualization of the training and testing curves for our method. As illustrated in Fig. 6, our approach demonstrates significantly faster convergence compared to models that are not pre-trained. This improved convergence highlights the effectiveness of our method in accelerating the learning process. Additionally, our method consistently outperforms the model without pre-training in terms of generalization. The performance on test sets remains superior throughout the entire training process, further underscoring the robustness of our method in handling unseen data and enhancing overall model accuracy. These results validate the benefits of incorporating pre-training, leading to more efficient and reliable model performance.

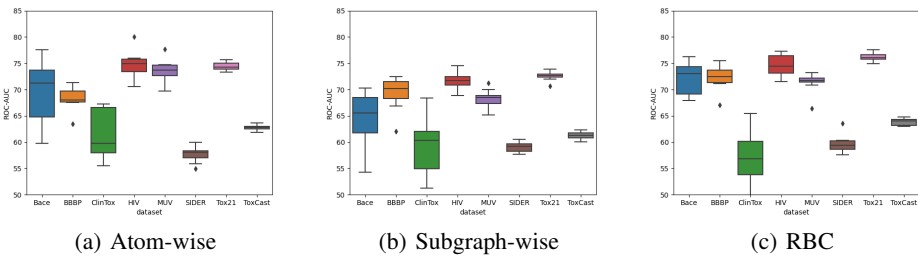

(a) Atom-wise           (b) Subgraph-wise           (c) RBC

Figure 7: Error bars for (a) the atom-wise branch, (b) the subgraph-wise branch, and (c) RBC.

# E   ERROR BARS

We also visualize the error bars in Fig. 7. For smaller datasets such as Bace and Clintox, there is significant uncertainty, as expected due to the limited data size. Notably, while the uncertainty in the subgraph-wise branch is slightly lower than that in the atom-wise branch, its performance remains inferior. Our method, which effectively fuses both atom-wise and subgraph-wise information, demonstrates the ability to explicitly reduce uncertainty while simultaneously improving performance. This balanced integration allows our approach to leverage the strengths of both branches, achieving better overall results despite the inherent challenges posed by smaller datasets.

