# OpenReview forum: "Integrating the Expression and Discrimination via Bilateral Compensation for Molecular Property Prediction"
_ICLR.cc/2025/Conference — ICLR 2025 Conference Withdrawn Submission_

### Official Review · Reviewer_A7x1 · 2024-10-24

**Soundness:** 2
**Presentation:** 3
**Contribution:** 2
**Rating:** 5
**Confidence:** 4

**Summary:**

This paper targets the molecular property prediction task and proposes a Route-guided Bilateral Compensation (RBC) architecture that extracts atom-wise and subgraph-wise information simultaneously. The authors also provide a theoretical generalization error bound for the proposed RBC. Additionally, a self-supervised learning task, MSTM, is proposed to jointly train the two branches of the model. The experimental results demonstrate the effectiveness of the proposed method compared to leading baselines in molecular property prediction tasks.

**Strengths:**

+ The paper is overall well-written, and the organization is good.
+ The proposed RBC architecture is interesting, and theoretical guarantees are provided.
+ The proposed method achieves good performance on several datasets.

**Weaknesses:**

+ The code to reproduce the experiments is not provided.
+ Some illustrations of RBC are vague and require further clarity.
+ Important baselines such as Graph Transformers [1] and Masked Graph Autoencoders [2] are missing. Additionally, experiments on large benchmark datasets [3] should be included.

[1] Do Transformers Really Perform Bad for Graph Representation?

[2] What's Behind the Mask: Understanding Masked Graph Modeling for Graph Autoencoders.

[3] Open Graph Benchmark. https://ogb.stanford.edu/

**Questions:**

+ RBC appears to be an advanced ensemble technique for representations with learnable weights. What is the technical novelty of RBC? How would it compare to a simple ensemble method for fusing the two modules?
+ It is suggested to list the dataset statistics either in the main content or in the appendix.
+ The authors claimed that they followed the dataset splits from [4]. Why is [4] mentioned but missing from the comparison?

[4] KPGT: Knowledge-guided pre-training of graph transformer for molecular property prediction.

---

### Official Review · Reviewer_tQS9 · 2024-10-30

**Soundness:** 2
**Presentation:** 3
**Contribution:** 2
**Rating:** 5
**Confidence:** 5

**Summary:**

This work targets the important molecular property prediction problem and introduce a Route-guided Bilateral Compensetaion architecture. Despite the charming theoretical analysis on its effectiveness, there are several issues that remain further addressed. For instance, the hierarchical molecular graphs and the MSTM, the two key components of this paper, have already been extensively explored in the past few years. However, the author accidently forgot them. Besides, the experimental results are not promising at the current stage, and the dataset is too small for verification.

**Strengths:**

(1) The figures are clean and readable. I can easily get the message.

(2) The theoretical analysis is solid. I like this part. Most existing methods do not have mathematical proofs of why their model works.

(3) The paper is well written. Sections are organized properly.

**Weaknesses:**

(1) Admittedly, molecular property prediction is a very challenging subfield in AI for molecular science, as there has already been a wide range of benchmarks. Despite the existing baselines listed in Table, many strong prior studies are not mentioned. See references below (I have skipped some smiles-based works, which may also be included for comparison):

A. Geometry-enhanced molecular representation learning for property prediction. Nature Machine Intelligence 2022.

B. Molecular contrastive learning of representations via graph neural networks. Nature Machine Intelligence. 2022

C. InstructBio: A Large-scale Semi-supervised Learning Paradigm for Biochemical Problems. 2023.

D. Self-supervised graph transformer on large-scale molecular data. Neurips 2020.

E. Molformer: Motif-based transformer on 3d heterogeneous molecular graphs. AAAI 2023.

F. One Transformer Can Understand Both 2D & 3D Molecular Data. ICLR 2023.

Besides, it is not correct that "other methods does not use knowledge or information other than molecular graphs". Liu et al. (2021) also use 3D geometry like 3DInfomax. I am concerned that authors are unable to clearly figure out the differences among those essential previous works.

i. Pre-training molecular graph representation with 3d geometry. ICLR 2022.

(2) MoleculeNet is acceptable as the benchmark dataset, but it is too old and many of its tasks have a very small number of samples. I strongly recommend the author examines their algoirthm over larger datasets, such as Molecule3D, QM9, PCBA, and PCQM4Mv2 from the OBG Large-scale Challenge. PCQM4Mv2 has over 3.37 million samples and can comprehensively test the effectiveness of their proprosed approach.

(3) The hierarchical modeling of molecular graphs is not novel to me [B-E]. The combination of atom-level graph and subgraphs have already been extensively discussed in many prior studies. More importantly, the self-supervised learning tasks are also not new. For instance, GROVE [A] proposes to mask motifs and require GNNs to classify the motif type, which is very similar to this work's MSTM. I highly recommend the author strictly compare their method with those papers and analyze the distinction.

[A] Self-Supervised Graph Transformer on Large-Scale Molecular Data. NeurIPS 2020.

[B] Molecular Representation Learning via Heterogeneous Motif Graph Neural Networks. ICML 2022

[C] HimGNN: a novel hierarchical molecular graph representation learning framework for property prediction. Briefings in Bioinformatics 2023

[D] Motif-aware Riemannian graph neural network with generative-contrastive learning. AAAI 2024

[E] Molformer: Motif-based transformer on 3d heterogeneous molecular graphs. AAAI 2023.


(4) The results in classification problems are not convincing. Those baselines are not competitive, and RBC still fails to achieve the SOTA over most tasks...

**Questions:**

(1) For subgraph extraction, the author mentions that predefined subgraph vocabularies are not ideal as not all molecules can be decomposed to disjoint subgraphs. This is not reasonable to me. Notably, it is not a necessity to decompose a molecule into disjoint subgraphs since some parts of this molecule may not be meaningful patterns.

Moreover, can the author explain more about the principle subgraph extraction? I found little information about how this mechanism is implemented. Actually, many prior works have explored this direction but are not clearly compared or discussed in the paper. See below:

[A] Molecular Representation Learning via Heterogeneous Motif Graph Neural Networks. ICML 2022

[B] HimGNN: a novel hierarchical molecular graph representation learning framework for property prediction. Briefings in Bioinformatics 2023

[C] Motif-aware riemannian graph neural network with generative-contrastive learning. AAAI 2024

[D] Molformer: Motif-based transformer on 3d heterogeneous molecular graphs. AAAI 2023.

(2) How did the author implement the graph augmentation for the graph-level self-supervised learning?


(3) In the experiments, the author conducts both regression and classification tasks, but in problem formulation, they said "we consider a binary classification problem"? Is this a typo or gap?

---

### Official Review · Reviewer_Uyfd · 2024-11-01

**Soundness:** 3
**Presentation:** 3
**Contribution:** 3
**Rating:** 5
**Confidence:** 4

**Summary:**

The authors introduce a Route-guided Bilateral Compensation (RBC) architecture that explicitly extracts atom-wise and subgraph-wise information through two decoupled branches, integrating these via a route module, with theoretical support provided. For different downstream tasks, the route module enables dynamic integration, enhancing the discriminative power of the final representation. External experiments confirm the effectiveness of the proposed method.

**Strengths:**

1. The authors conduct extensive experiments to demonstrate the effectiveness of the proposed method.

2. The theoretical proof is particularly interesting.

**Weaknesses:**

1. The Related Work section should be further strengthened to clarify the differences between this work and the latest research in the field. This would help position the proposed method more clearly within the context of recent advancements.


2.  Limitation Section is not included.

**Questions:**

1. The authors claim that "most existing self-supervised molecular learning methods, primarily designed for atom-based architectures, cannot fully capture subgraph-wise information and the relations among substructures." However, recent studies such as "Atomic and Subgraph-aware Bilateral Aggregation for Molecular Representation Learning" , "Chemistry-Wise Augmentations for Molecule Graph Self-supervised Representation Learning" ，“Junction Tree Variational Autoencoder for Molecular Graph Generation  ”and "A Comprehensive Study on Large-Scale Graph Training: Benchmarking and Rethinking" appear to address this aspect.


2. It would be beneficial to consider incorporating more diverse approaches for substructure division, as this step seems crucial to performance and may significantly impact results.


3. The overall clarity of the paper could be improved for better readability.


4. A discussion on the limitations of the proposed method would strengthen the paper, providing a balanced perspective.

---

### Official Review · Reviewer_9Vhb · 2024-11-02

**Soundness:** 2
**Presentation:** 3
**Contribution:** 2
**Rating:** 3
**Confidence:** 5

**Summary:**

This paper introduces a novel architecture called Route-guided Bilateral Compensation (RBC) for molecular property prediction, which addresses limitations in traditional approaches by incorporating both atom-wise and subgraph-wise branches. The RBC model dynamically fuses these two types of information based on the nature of molecular properties, allowing for adaptive feature selection between atoms and subgraphs. The authors provide theoretical analysis showing that the subgraph-wise branch has higher expressiveness and demonstrate through empirical results that integrating both branches leads to improved generalization. Additionally, the model includes a unique self-supervised learning strategy, including Masked Subgraph-Token Modeling (MSTM) for subgraphs, to further enhance the learning process.

**Strengths:**

①The paper makes a contribution by decoupling atom-wise and subgraph-wise information into separate branches within the RBC model, enabling the dynamic selection of relevant features.

②The proposed MSTM for subgraph-level learning in combination with existing atom-level tasks is noteworthy. The design ensures that RBC captures multi-level information, from individual atoms to larger functional groups, thereby supporting robust and transferable molecular representations across tasks.

**Weaknesses:**

①Limited Comparative Analysis: While there are numerous innovative methods for molecular graph representation learning based on atom-level and subgraph-level information, the authors have selectively ignored these existing approaches (e.q., MolCLR[1]) in their experiments. Instead, they only compared their method with more general self-supervised techniques from previous years, which undermines the demonstration of their approach's effectiveness. Notably, the performance, especially in molecular property classification tasks, does not significantly surpass these general methods. In fact, it fails to outperform them on individual datasets, only achieving a slight edge in average performance.

1.Wang Y, Wang J, Cao Z, et al. Molecular contrastive learning of representations via graph neural networks[J]. Nature Machine Intelligence, 2022, 4(3): 279-287.

②Unclear Theoretical Analysis: The theoretical analysis presented in Section 3.5 does not effectively convey the authors' intended message. Although the theoretical derivation of the generalization bound suggests potential advantages of the fusion approach, the specific impacts of different components of the generalization error—such as the covariance term—on practical tasks are not intuitively clear. Furthermore, the discussion on the dynamic selection of weight λ lacks sufficient intuitive explanations or illustrative examples, making the theoretical foundations difficult to understand. Generally, the theoretical bounds serve as upper estimates of model performance, which can be influenced by dataset distribution and noise. The absence of a clear connection between theory and experimental results leaves readers uncertain about the practical applicability of these insights.

③Insufficient Evidence of Effectiveness: The combination of ambiguous theoretical claims and modest performance improvements does not convincingly validate the proposed method's effectiveness or substantiate the superiority of its theoretical foundations.

**Questions:**

①Given that the authors did not compare their approach with recent atom-level and subgraph-level representation learning methods, could they explain their rationale for focusing solely on more general self-supervised methods? How do they believe this choice impacts the perceived effectiveness of their model?

②In Section 3.5, the theoretical analysis seems to lack intuitive clarity, particularly regarding the dynamic selection of weight λ. Could the authors provide more concrete examples or a detailed explanation of how this weight selection process influences the model's performance in practice?

---

### Official Review · Reviewer_Q28h · 2024-11-04

**Soundness:** 2
**Presentation:** 2
**Contribution:** 2
**Rating:** 3
**Confidence:** 3

**Summary:**

This paper addresses the challenge of predicting molecular properties by proposing a novel Route-guided Bilateral Compensation (RBC) architecture. This approach aims to integrate atom-wise and subgraph-wise information, which are crucial for accurately capturing molecular characteristics. Extensive empirical evaluations shows that the learned representation has a stronger generalization ability in various functional group-related molecular property prediction tasks.

**Strengths:**

- Presentation of the proposed method and its preliminaries are clear, the main contribution of this work can be clearly understood.
- Experiments can effectively reflect the intended objectives of the model.

**Weaknesses:**

- Novelty of the proposed method. The idea of RBC is to mix the atom-wise embedding and subgraph-wise embedding with a learnable parameter, which is commonly used in mix-up methods.
- The theoretical proof is misleading. For example, Theorem 1 is not closely related to the model proposed in this work. In addition, the authors only used the WL test on an example as a proof of Theorem 1, which is completely wrong. It should be justified mathematically.
- The experiment lacks hyperparameter sensitivity analysis.

**Questions:**

What is the distribution of \lambda across the molecular dataset?

---

### Note · Authors · 2025-01-15

I have read and agree with the venue's withdrawal policy on behalf of myself and my co-authors.